# OpenReview forum: "Biases in the Blind Spot: Detecting What LLMs Fail to Mention"
_ICML.cc/2026/Conference — ICML 2026 regular_

### Official Review · Reviewer_WgYm · 2026-02-23

**Soundness:** 3
**Presentation:** 3
**Significance:** 3
**Originality:** 3
**Overall Recommendation:** 4
**Confidence:** 3

**Summary:**

This paper introduces a fully automated, black-box pipeline for detecting unverbalized biases in LLMs, where systematic decision factors that influence model outputs but are never cited as justification in cot reasoning. The pipeline operates in multiple stages: (1) clustering task inputs and using an LLM autorater to generate candidate bias concepts from representative samples; (2) generating positive/negative counterfactual input variations via another LLM; (3) filtering variations with an LLM quality judge; (4) collecting target model responses and checking verbalization rates with yet another LLM; and (5) applying McNemar's test with Bonferroni correction and O'Brien-Fleming early stopping for statistical significance. The authors evaluate across three binary decision tasks (hiring, loan approval, university admissions) on six LLMs, rediscovering known biases (gender, race) and uncovering novel ones (Spanish fluency, English proficiency, writing formality). Ablation studies include seed consistency, intentional bias injection (92.5% accuracy), verbalization detector validation (κ > 0.6 for all tested detectors), and comparisons with four prior bias studies.

**Compliance With Llm Reviewing Policy:**

Affirmed.

**Key Questions For Authors:**

Please see Weakness part.

**Limitations:**

yes

**Strengths And Weaknesses:**

## Strength
1. The motivation of paper is pretty attractive, where LLMs may hide what they think to affect the final decision make and causing the bias.

2. I think the statistical methods makes experiment rigorious, where author uses McNemar's test for paired outcomes, Bonferroni FWER control, O'Brien-Fleming staged testing, and futility stopping are all appropriate choices.

## Weakness
1. I am concerned that if detected concepts are atomic, which means that each variation may bundles multiple signals, and we don't know which one the model responds to. The pipeline tests high-level concepts like "Spanish fluency", but this kind of variation may carry multiple entangled signals, for example: "Fluent in Spanish" = language skill + possible Hispanic identity. If "Fluent in French" produces the same effect, the model just values multilingualism. If only Spanish works, there is an ethnic component. The paper never tests this.

2. For Verbalization Detection Evaluation, why do you choose to use GPT-5-mini as judge instead of GPT-4.1-mini who got best performance overall.

3. One small suggestion. Author uses almost 1.5 page on related works, limitations and conclusion, I suggest to reconstruct the paper where move more experiment details from appendix to main pages.

I pretty love your motivation and topic, if author could address my concerns, I would consider to raise my score.

---

> ### Author Rebuttal · Authors · 2026-03-31
>
> We ran a multilingual control ablation study testing French, German, and Mandarin fluency alongside Spanish on four models, directly addressing the concept atomicity concern. We also address the concerns about the model choice for verbalization detection.
>
> We hope these results address the reviewer's concerns and respectfully ask whether they would consider updating their score. We thank the reviewer for noting the "attractive" motivation and that the statistical methods "make the experiment rigorous."
>
> **W1: Concept atomicity.** In general, our pipeline does not claim to fully disambiguate all possible reasons behind any detected unverbalized bias. That being said, we agree that "Spanish fluency" could bundle language skill with Hispanic identity, and that disentangling these signals is an interesting question worth studying. We have run targeted follow-up experiments to address this directly.
>
> We tested fluency in French, German, and Mandarin alongside Spanish fluency, using the same counterfactual generation pipeline on the hiring task across four models. Results at the initial screening stage (~200 inputs per concept):
>
> | Language | gemma-3-12b | gemma-3-27b | gemini-2.5-flash | claude-sonnet-4 |
> |----------|-------------|-------------|------------------|-----------------|
> | Spanish | +5.0% (p=.031) | +4.5% (p=.049) | +3.5% (p=.143) | +4.5% (p=.035) |
> | French | +1.0% (p=.804) | -0.5% (p=1.00) | +1.5% (p=.629) | +2.0% (p=.388) |
> | German | +1.0% (p=.774) | +1.0% (p=.791) | -2.5% (p=.332) | +0.0% (p=1.00) |
> | Mandarin | +5.5% (p=.013) | +0.0% (p=1.00) | +2.5% (p=.332) | +1.5% (p=.508) |
>
> Spanish fluency consistently shows the largest effect across models. French and German show negligible effects (delta < 2%, p > 0.6). Mandarin shows a strong effect on gemma-3-12b (+5.5%, p=.013) but not on others. This pattern suggests that the bias has an ethnic or cultural component beyond general multilingualism, since a purely multilingualism-based bias would affect all languages similarly. One possible explanation is that Spanish and Mandarin are more strongly associated with racial/ethnic minority groups in a US context than French or German, but we cannot confirm this from the current data alone.
>
> **W2: GPT-5-mini vs GPT-4.1-mini for verbalization detection.** GPT-5-mini was chosen because it is more recent and cheaper for input tokens, which is the bottleneck for verbalization detection (the task is input-heavy: each response must be read in full to check whether a concept is mentioned). GPT-4.1-mini has higher per-token input cost. The two models disagree on only 8% of samples in our evaluation set (Appendix K), so the improvement was not significant enough to justify switching.
>
> **W3: Paper structure.** We agree and appreciate the suggestion. We have trimmed related work, limitations and conclusion in the paper to add space for more experimental details.

---

> > ### Author Rebuttal · Reviewer_WgYm · 2026-03-31
> >
> > Authors address my concerns, I would raise my score to 4

---

> > > ### Author Response · Authors · 2026-04-01
> > >
> > > Thank you for engaging with our rebuttal and for raising your score. We are glad the multilingual control experiment addressed the atomicity concern.
> > >
> > > Since you noted that your concerns have been "fully resolved," would you consider raising your score to accept? We believe the paper makes a solid contribution and would appreciate your full support.

---

### Official Review · Reviewer_bnRL · 2026-03-10

**Soundness:** 3
**Presentation:** 3
**Significance:** 1
**Originality:** 2
**Overall Recommendation:** 4
**Confidence:** 4

**Summary:**

The paper studies whether LLMs make decisions (hiring, loan, or admissions) based on irrelevant demographic information included in the prompt. The authors study whether LLMs are biased by examining if they make systematically different decisions based on (1) the mere inclusion of demographic information, and (2) the nature of that inclusion (i.e., enhancement of a given concept). They contextualize their findings through the lens of faithfulness of verbalization: does the chain-of-thought (CoT) produced alongside a decision faithfully reflect the decision making process, particularly when a model arrives at different decisions depending on demographic inclusions? They term this phenomenon "unverbalized biases".

The authors introduce an automated LLM-based framework that: (1) takes text samples (without demographic information), embeds them in a text embedding model, (2) uses K-means clustering to identify similar pieces of text, and (3) from each cluster samples a small number of examples to prompt an LLM to generate concepts, and (4) eventually generate a set of positive and negative samples that enhance or diminish the presence of a given concept. Rather than exhaustively searching for concepts or sample pools for a given concept, they use an iterative elimination-based algorithm to identify concept significance using McNemar's test statistic and verbalization rate (also determined by an LLM), with stopping criteria based on O'Brien-Fleming boundaries and futility thresholds computed via Monte Carlo simulation. They find evidence of known biases (gender, race) as well as previously unreported biases (Spanish/English proficiency and writing formality).

**Compliance With Llm Reviewing Policy:**

Affirmed.

**Key Questions For Authors:**

1. Why do the you think that you observe biases that prefer marginalized groups? Can you hypothesize what about the models, their formulation / training might be causing this unintuitive behavior? Have you studied stereotypical associations? For instance, in the hiring task, are there certain occupations for which women are preferred over men (in the style of WinoBias, which studies occupational stereotypes)? Is there stereotypical alignment in the biases that would make your results more interpretable? Are the occupations in your study predominantly female-coded (e.g., nurse)?

2. What would the relationship between tau and unverbalized bias be?

3. Contributions 1 and 2, as stated, seem to point to a single contribution: using LLMs to study unverbalized biases. Can you clarify, concretely, how they are distinct?

4. Does your measure of unverbalized bias account for both explicit and implicit forms of verbalization? For instance, explicitly mentioning a demographic attribute (e.g., "the candidate is female") versus implicitly referencing a known proxy (e.g., mentioning traits or language statistically associated with a demographic group)?

5. The use of LLMs to generate concepts does not guarantee coverage of all unknown concepts. The fact that the majority of the identified concepts for unverbalized bias (with the exception of three that are significant and not gender- or race-related) are already known biases suggests that LLMs may not sufficiently understand the socio-technical context to provide adequate coverage of relevant + significant concepts. Why are LLMs and black-box solutions a suitable path forward here, as opposed to (or in conjunction with) multi-disciplinary / participatory approaches?

**Limitations:**

Yes.

**Strengths And Weaknesses:**

*Strengths*

1. The design of the pipeline in Algorithm 1 appears sound.
2. The design of experiments is thorough. Ablations are careful; for instance, studying the effect of random seed ensures that results are not simply an artifact of temperature or experimental stochasticity.
3. The paper is clearly written and well structured. The appendix is thorough, and the experimental claims are well supported by the findings.
4. Transparent reporting of cost. This is much appreciated!

*Major weakness*
Even though the experiments / methods are sound, the following weakness significantly undermines the motivation (and therefore the projected impact / significance) for this work. The authors' core motivation for this work is collectively that: (1) CoT is actively being used as a tool in monitoring production models, and (2) when CoT is not faithful to its thinking or decision making, it limits institutional ability to monitor bias in production systems. In practice through, production systems are typically assessed and monitored using behavior-based datasets (e.g., StereoSet, BBQ), or through methods like red teaming, or output harm/bias/toxicity classifiers. Therefore, the motivation for this work is weak. One of the stated expected impact of this work is to study bias beyond stated reasoning, but it is already standard practice in industry to look beyond stated reasoning for bias detection / monitoring.

*Minor weaknesses that did not affect my review / score*
1. It appears that phi and z are not fully defined at line 194.
2. The example in Figure 1 appears to be a failure case for the variation generator. While it is plausible for a "Miguel Garcia" to be a practicing Hindu, it is unlikely, and there is potential for these kinds of errors to confound the analysis.
3. The use of the word "efficient" in line 102 seems overstated.

---

> ### Author Rebuttal · Authors · 2026-03-31
>
> We provide new ablation studies as suggested: pro-marginalized bias patterns (22:0 pro-female, 17:0 pro-minority across all models) and tau threshold sensitivity showing 70% of biases are deeply hidden (verbalization rate <5%). We also address the motivation concern and all questions below.
>
> We hope these results address the reviewer's concerns and would appreciate it if the reviewer considered strengthening their support. We thank the reviewer for noting the "sound" pipeline design, "thorough" experimental design, "careful" ablations, and "transparent reporting of cost."
>
> **Major weakness: Motivation.** We respectfully disagree that the motivation is significantly undermined.
>
> Firstly, StereoSet (stereotypical word associations) and BBQ (stereotype-driven QA errors along predefined categories) both require knowing what to test for. Our pipeline addresses a complementary problem: **discovering unknown biases without predefined categories.** Writing formality and language proficiency do not appear in any standard benchmark. Output harm/toxicity classifiers also cannot catch unverbalized biases specifically, since by definition the problematic factor never appears in the model's output.
>
> In addition, general benchmarks like StereoSet and BBQ are also not task-specific. Our empirical results show that the direction of **biases in high-stakes decisions often contradicts what benchmarks like StereoSet would surface**: we observe pro-marginalized preferences that run counter to the stereotypical direction. This shows that task-specific automated discovery is necessary.
>
> Finally, the unverbalized filter has specific value for practitioners who use CoT monitoring for safety oversight (cf. Anthropic's responsible scaling policy, OpenAI's recent work on CoT monitoring), or for auditing the reasons behind decisions (required for many fields). Our filter identifies biases invisible to this layer. For practitioners who do not use CoT monitoring at all, behavioral testing suffices, but our primary audience are those building AI safety oversight or auditing systems where the stated reasoning matters.
>
> **Q1: Pro-marginalized biases.** Across all tasks and models, we observe **22 pro-female vs 0 pro-male** and **17 pro-minority vs 0 pro-majority** significant unverbalized biases. Every model that shows gender or race bias favors the marginalized group. This likely reflects overcorrection from RLHF/RLAIF safety training. The effect is consistent across model families (Gemma, Gemini, GPT-4.1, Claude, QwQ) rather than increasing with model recency, suggesting it is a widespread consequence of current alignment practices.
>
> **Q1 (cont.): Stereotypical associations.** Our hiring evaluation uses the software engineering job description from [1]. A WinoBias-style analysis varying the target occupation (e.g., nurse vs. engineer vs. CEO) to test whether gender bias direction aligns with occupational stereotypes is a valuable extension and we now discuss it in the paper.
>
> **Q2: Tau-bias relationship.** Tau controls filtering stringency: concepts verbalized in more than tau fraction of responses are excluded. In the hiring task, the distribution of verbalization rates among significant concepts is heavily skewed toward zero: **70% of significant concept-model pairs have verbalization rate below 0.05**, meaning the model almost never mentions these factors. At tau=0 (only concepts never mentioned in any response), ~25-30% of significant concepts survive across models. At tau=0.05, ~65-75% survive. At tau=0.10, ~77-83% survive. At the paper's default tau=0.3, all significant concepts pass. The vast majority of unverbalized biases are deeply hidden, not borderline cases near the threshold.
>
> **Q3: Contributions 1 vs 2.** Fair point, we have combined these into a single bullet in the paper.
>
> **Q4: Implicit vs explicit verbalization.** Our detector uses an LLM judge to check whether the concept is explicitly cited as a decision factor in the model's reasoning. Implicit proxies (traits statistically correlated with a demographic group without being referenced) are not considered verbalized under the current implementation, but the verbalization detector can be modified to include this if desired. We have added a discussion of this to the paper.
>
> **Q5: Concept coverage.** We agree LLM-generated concepts do not guarantee full coverage and view automated and human-in-the-loop approaches as complementary. The pipeline architecture supports injecting human-proposed concepts alongside LLM-generated ones. In fact, this is how we ran some of our ablation studies using explicitly chosen concepts.
>
> **Minor weaknesses.** We have defined phi and z at line 194, acknowledged the Figure 1 variation generator limitation (Miguel Garcia as Hindu), and replaced "efficient" with more precise language about sequential testing's cost reduction.
>
> [1] Karvonen & Marks, 2025, Robustly Improving LLM Fairness in Realistic Settings via Interpretability

---

> > ### Author Rebuttal · Reviewer_bnRL · 2026-04-03
> >
> > I thank the authors for their responses, which have resolved the majority of my concerns. However, my concern regarding concept coverage remains. I do not think this can be fully addressed within the rebuttal, and I would recommend that the authors acknowledge this as a limitation in the final version of the paper. Given that my other concerns have been satisfactorily addressed, I am raising my score accordingly.

---

> > > ### Author Response · Authors · 2026-04-04
> > >
> > > Thank you for engaging with our rebuttal and for indicating that you are raising your score. We appreciate your careful review.
> > >
> > > **Concept coverage as a limitation.**
> > >
> > > We agree this is a genuine limitation. We have updated the “Concept Hypothesis Coverage” subsection of “Limitations” to explicitly acknowledge the point you raise: because our pipeline can only detect biases that the hypothesis-generation LLM proposes, it inherits whatever blind spots that model has about the socio-technical context of the task. We also added discussion of a partial mitigation by using a different LLM for hypothesis generation than the model being analyzed, ideally one with different biases, so that blind spots are less likely to be shared between the generator and the target. We further note that automated discovery is best viewed as complementary to, not a replacement for, multi-disciplinary and participatory approaches, and that combining LLM-generated hypotheses with concepts proposed by domain experts and affected communities is a necessary direction for future work.
> > >
> > > **Score update.**
> > >
> > > We noticed that the score on the submission does not yet appear to reflect the increase you mentioned. If you have a moment, we would be grateful if you could update the score directly so the change is reflected in the system. Thank you again for the constructive engagement.

---

### Official Review · Reviewer_poQg · 2026-03-10

**Soundness:** 3
**Presentation:** 1
**Significance:** 3
**Originality:** 2
**Overall Recommendation:** 4
**Confidence:** 3

**Summary:**

This paper presents a llm-based automated pipeline that detects LLMs hidden biases (biases that systematically influence model decisions in high-stakes tasks but are never mentioned in the model's CoT reasoning.) The pipeline works by using LLMs to generate candidate bias concepts, create counterfactual input variations, and apply statistical testing to identify significant unverbalized effects.

**Compliance With Llm Reviewing Policy:**

Affirmed.

**Final Justification:**

Most of my concerns have been addressed. I still feel the overall structure of the paper and the presentation of the conclusions could be stronger, but as the authors have promised to address this in the revision, I raised my score.

**Key Questions For Authors:**

Please see main limitations anove.
The three tasks evaluated (hiring, loan approval, university admissions) share a similar structure: a single binary decision made over a structured input profile. It would be helpful if the authors discussed whether the pipeline is expected to generalize to less structured tasks, such as open-ended question answering or multi-turn dialogue, and what modifications might be needed.

**Limitations:**

yes

**Strengths And Weaknesses:**

I appreciate the great amount of experiments performed in this paper, and the results appear rigorous. However, the main text needs revisions: it spends too much space on background and experimental setup while neglecting to present the results in a way that sufficiently highlights the paper's core contributions. Specifically:


- **The current presentation of results is unintuitive.** Tables 1, 2, and 3 contain a large number of "–", which according to the captions can mean two things: either it was `not flagged as significant`, or it was `filtered due to verbalization`. These two cases have very different implications, yet they are indistinguishable in the tables. This can confuse readers, as it is unclear whether a "–" entry means the model has no such bias, or that the bias exists but was openly stated in the model's reasoning. Differentiate between these two cases in the tables would be helpful.

- **Bias effect discussion** The detected bias effect are generally in the range of 2–5%. And authors acknowledge in Appendix 3 that these effects are much smaller than those found in human labor market discrimination studies (e.g., the 50% callback rate gap). So they do not sufficiently discuss whether a 2–5% effect constitutes meaningful harm or merely reflects model randomness. It would also strengthen the paper if the authors considered finding scenarios where bias effects are more pronounced, and presented results in a more intuitive way to better support the paper's core claims.

- **Unified sensitive features.** The paper does not systematically present the same set of sensitive features across all three tasks, making cross-task comparison difficult. For example, Table 1 reports biases under Gender, Race, and Other (e.g., Spanish language ability); Table 2 reports Gender, Race, and Language/Tone (e.g., English proficiency); and Table 3 only reports Gender and Race. This inconsistency raises questions that the current results cannot answer — for instance, does language proficiency bias also exist in the hiring task? It would strengthen the paper if the authors defined a consistent set of sensitive features to be tested across all three tasks, with any additional task-specific findings reported in the appendix.

---

> ### Author Rebuttal · Authors · 2026-03-31
>
> We revised the tables to distinguish "not significant" from "verbalized" (83% of "--" entries are verbalized, not absent), ran cross-task ablation studies testing unverbalized concepts, and address effect size and generalization concerns below.
>
> We hope these results address the reviewer's concerns and respectfully ask whether they would consider updating their score. We thank the reviewer for acknowledging the rigor and extent of our experiments.
>
> **W1: Table presentation.** We agree that conflating "not significant" and "filtered by verbalization" under a single "--" is confusing, and we appreciate this specific feedback. These two cases have very different implications: one means the concept was tested but showed no significant effect, and the other means a significant causal effect *was* found, but was excluded because the model explicitly references the concept in its CoT reasoning. The revised tables use distinct notation (e.g., "n.s." for not significant, "verb." for filtered by verbalization). To provide immediate clarity on the current tables:
>
> Across all tasks and models, only **17% of "--" entries are "not significant"** while **83% are "filtered by verbalization"** (the model mentions the concept in its CoT, so the bias is not hidden). For example, in Table 1 (hiring), gemma-3-12b-it has 89 "--" entries: 26 are not significant and 63 were filtered because the model verbalizes the concept. This breakdown itself reinforces our core finding: the majority of tested concepts do produce significant biases, but most are verbalized. The unverbalized biases we report are the small, specifically concerning subset that evades CoT monitoring.
>
> **W2: Bias effect sizes.** The bias effects found in the paper are statistically significant effects after Bonferroni correction, not model randomness. The pipeline's sequential testing with futility stopping specifically distinguishes signal from noise. As mentioned in the paper, they also match similar effect sizes from other works (Karvonen & Marks, 2025, "Robustly Improving LLM Fairness in Realistic Settings via Interpretability").
>
> Whether a detected factor is inappropriate or harmful for a given task is context-dependent
> and requires task-specific audit. However, systematic effect sizes of 2-5% can affect tens of thousands of outcomes when AI systems are deployed at scale, especially when taking into account agentic or multi-turn setups, where effects can compound. Our conservative statistical design (Bonferroni, O'Brien-Fleming, quality filtering), along with our experiment parameters and threshold, means reported effects are lower bounds.
>
> **W3: Unified features.** We agree that different feature sets across tasks limits cross-task comparison. We ran follow-up experiments testing novel concepts from one task in the other two (gemma-3-12b-it):
>
> - *English proficiency* (significant unverbalized in loans): produced a large significant effect in hiring (+21.5%, p<0.001), but the model *verbalizes* it in hiring (mentioned in 79% of CoT responses). Verbalized in university admissions as well.
> - *Writing formality* (significant unverbalized in loans): showed a small negative trend in hiring (delta=-1.3% to -5.5% across stages), not reaching significance. The effect does not generalize from loans to hiring.
> - *Spanish fluency* (significant unverbalized in hiring on qwq-32b): not significant in loan approval (delta=-0.5%, p=0.77). In university admissions, showed a positive trend (+4.8%) but did not reach significance and was verbalized (54% rate).
>
> The results are mixed. English proficiency bias generalizes across tasks but whether it is *unverbalized* depends on the task: hidden in loan decisions, but stated openly in hiring reasoning. Writing formality and Spanish fluency appear task-specific. This is why **automated task-specific discovery matters**: both the existence and the verbalization status of biases vary by context. Unified cross-task tables are now in the paper.
>
> **Q1: Generalization.** We chose binary decision tasks not because the pipeline requires them, but because they provide a clean, unambiguous unbiased baseline: a fair system should accept equally-qualified candidates at roughly equal rates regardless of the concept being varied. This makes deviations from baseline directly interpretable as bias.
>
> The **pipeline is not fundamentally restricted to binary decisions.** For open-ended tasks, one replaces the binary accept/reject label with any appropriate outcome measure (e.g., a quality judge scoring responses on a scale, a toxicity classifier, word-count statistics on mentions of specific attributes). The counterfactual structure (generate paired inputs that differ only in one concept, measure behavioral difference) applies unchanged. The revised manuscript expands on this discussion.

---

> > ### Author Rebuttal · Reviewer_poQg · 2026-04-04
> >
> > Thanks for the response, most of my concerns have been addressed. I still feel the overall structure of the paper and the presentation of the conclusions could be stronger, but as the authors have promised to address this in the revision, I'll raise my score.

---

> > > ### Author Response · Authors · 2026-04-06
> > >
> > > Thank you for engaging with our rebuttal and for raising your score. We appreciate the constructive feedback throughout.
> > >
> > > We have revised the table notation (distinguishing "n.s." from "verb."), added unified cross-task comparison tables, and restructured the results presentation as discussed. We want to make sure these changes fully address the presentation concerns you raised.
> > >
> > > If you have specific suggestions for how the conclusions could be strengthened further, we would welcome them and are happy to incorporate changes in the camera-ready. If the revisions we have described adequately address the remaining concerns, we would be grateful if you considered whether the paper merits stronger support.

---

### Official Review · Reviewer_JEtR · 2026-03-13

**Soundness:** 3
**Presentation:** 4
**Significance:** 2
**Originality:** 2
**Overall Recommendation:** 4
**Confidence:** 4

**Summary:**

The authors examine unverbalized biases in Large Language Models (LLMs)—biases that influence a model's final decision but are strategically or implicitly omitted from its chain-of-thought (CoT) reasoning. The paper proposes an automated black-box pipeline that uses LLM-based clustering and brainstorming to identify task-specific bias concepts, generates counterfactual input pairs to test for causal influence, and applies statistical significance tests (McNemar’s test with O’Brien-Fleming alpha spending) to detect biases while controlling API costs. The pipeline specifically filters for biases that do not appear in the model's self-stated reasoning. The general aspect of this work is promising, as it identifies both known demographic biases and novel factors such as writing formality and language proficiency.

**Compliance With Llm Reviewing Policy:**

Affirmed.

**Key Questions For Authors:**

1. Given that standard significance testing on outputs will catch both verbalized and unverbalized biases, can you further justify why the distinction of "unverbalized" is a major problem for practitioners compared to general bias detection?

2. Have you conducted sensitivity analyses by running the generative stages (2 and 3) under different random seeds?

3. How does the brainstorming/clustering approach (Stage 1) specifically benefit from the "unverbalized" prompt constraint compared to a general bias brainstorming session?

**Limitations:**

yes

**Strengths And Weaknesses:**

**Strengths:**

- **Discovery of Novel and Known Biases:** The pipeline successfully rediscovers biases identified in prior manual research (such as gender and race) while also measuring biases like writing formality and language proficiency.

- **Methodological Soundness:** The pipeline is technically well-constructed, particularly in its use of sequential testing and early stopping to manage the high costs of API-based evaluation. This makes the large-scale testing of many concepts computationally feasible. Also, meta-evaluation is done by hand on the validity of LLM pipelines, strengthening its soundness.

- **Extensive Evaluation:** The authors evaluate their framework across many models and realistic, large-scale datasets providing a broad empirical base for their findings.

- **Clarity of Presentation:** The paper is well-organized and provides a clear narrative regarding the steps of the pipeline, and the inclusion of detailed prompts in the appendix supports reproducibility.

**Weaknesses:**

- **Conceptual Significance of "Unverbalized":** A central claim of the paper is the importance of detecting _unverbalized_ bias. However, since the pipeline **ultimately** relies on standard causal significance testing with no regard to the chain-of-thought outputs, any significant bias would be caught by these tests. Given that significance testing is a standard scientific tool, the framing of "unverbalized" as a unique problem may be overstated; a researcher performing standard bias audits would likely discover these same biases regardless of whether the model mentions them in its CoT.

- **Originality Concerns:** The individual components, counterfactual generation via LLMs and automated bias discovery, build heavily on existing paradigms. The novelty lies primarily in the combination and the specific filtering for CoT omission, but as noted, the practical value of this filter is debatable if the goal is general bias mitigation. For instance, extending counterfactual faithfulness testing via templates is explored in prior work (e.g., Atanasova et al., 2023). Additionally, while the clustering approach for concept generation is utilized, it would strengthen the paper to discuss how this compares to other automated bias discovery techniques in the broader literature (such as teacher-student seed expansion methods like BiasScope) to clarify the specific advantages of clustering.

- **Potential for Confounds:** While the authors discuss variation quality, there is a risk that the LLM variator introduces hidden perturbations. The robustness of the counterfactual generation (Stages 2 and 3) is not fully explored across multiple seeds, which is critical for ensuring that the detected "bias" is not an artifact of the specific counterfactuals generated.

---

> ### Author Rebuttal · Authors · 2026-03-31
>
> We ran a seed sensitivity analysis confirming variation robustness (0.64pp std across 3 seeds at temperature=0), and provide clarifications on the unverbalized motivation, concept generation prompt, and comparison with prior & concurrent work.
>
> We hope these results address the reviewer's concerns and would appreciate it if the reviewer considered strengthening their support. We thank the reviewer for noting the pipeline's "methodological soundness," the "extensive evaluation" across models, and the "clarity of presentation."
>
> **W1/Q1: Why "unverbalized" matters for practitioners.** We agree that, given a predefined set of bias concepts, standard significance testing catches biases regardless of verbalization. The distinction matters at the stage *before* testing: where do the concepts come from?
>
> Existing bias audits answer the question "is the model biased along dimension X?" Our pipeline answers a harder, prior question: "along *which* dimensions is the model biased?" Behavioral benchmarks (StereoSet, BBQ) and compliance frameworks all require researchers to specify which biases to test for. Our pipeline discovers candidate concepts from the data, without predefined categories. Writing formality and language proficiency do not appear in any existing benchmark, and a researcher performing standard audits would not test for these without prior reason to suspect them.
>
> In addition, the unverbalized filter has specific value for practitioners who use CoT monitoring for safety oversight (cf. Anthropic's responsible scaling policy, OpenAI's recent work on CoT monitoring), or for auditing the reasons behind decisions (required for many fields). Our filter identifies biases invisible to this layer. For practitioners who do not use CoT monitoring at all, behavioral testing suffices, but our primary audience are those building AI safety oversight or auditing systems where the stated reasoning matters.
>
> We also note that benchmarks like StereoSet and BBQ are not task-specific. Our empirical results show that the biases found in high-stakes decisions often run counter to the direction those benchmarks would predict (notably, the pro-marginalized preferences we observe).
>
> **Q3: Brainstorming constraint.** The concept generation prompt is a modular component of the pipeline, not a fixed design choice. Changing it will produce different candidate concepts, and the prompt can be tailored to the practitioner's needs. The downstream stages (variation generation, verbalization filtering, statistical testing) are what validate whether a candidate concept is a real unverbalized bias, regardless of how it was generated.
>
> **W3/Q2: Seed sensitivity.** We re-ran variation generation with 3 different random seeds for 9 concepts from the hiring dataset (4 significant unverbalized, 5 non-significant), using gemma-3-12b-it at temperature=0 to isolate variation generation stochasticity from model stochasticity. Across all 9 concepts, the mean standard deviation in bias estimates across seeds was *0.64 percentage points* (significant concepts: 0.75 pp, non-significant: 0.56 pp). Bias direction was consistent across all seeds for every concept. This confirms that detected biases are robust to the specific counterfactuals generated and are not artifacts of particular variation wordings.
>
> **W2: Comparison with prior work.** The reviewer also mentions Atanasova et al. (2023). We already cite this as a methodological ancestor in our contributions: our pipeline extends their counterfactual faithfulness testing in three ways: (1) LLM-based concept variation replaces their per-task trained T5 editor, removing the need for task-specific training; (2) semantic verbalization checking replaces syntactic token overlap; (3) we add statistical testing (McNemar's + Bonferroni) where they report raw percentages.
>
> Regarding BiasScope, we believe the reviewer refers to Lai et al. (2026, "BiasScope: Towards Automated Detection of Bias in LLM-as-a-Judge Evaluation"), concurrent work after our submission. BiasScope discovers biases in LLM-as-a-Judge pairwise evaluation by starting from 7 seed bias definitions (length, position, authority, etc.), injecting them into responses via a teacher LLM, analyzing the target model's failures to identify new biases, and iterating. The main differences worth noting are: (1) BiasScope requires predefined seed biases to bootstrap discovery; our pipeline requires none, generating concepts directly from task inputs via clustering and brainstorming. (2) BiasScope perturbs the *response* to test whether the judge is swayed; we perturb the *input* to test whether the model's decision changes. (3) BiasScope validates with a single directional error rate comparison; we use statistical tests. (4) BiasScope's iterative expansion loop, where failures under known biases reveal new ones, has no analogue in our single-pass pipeline. We view the approaches as complementary and have added this discussion to the paper.

---

> > ### Author Rebuttal · Reviewer_JEtR · 2026-04-03
> >
> > I have read the authors' rebuttal and appreciate the detailed clarifications and the additional experiments conducted during the rebuttal period.
> >
> > First, I want to thank the authors for running the seed sensitivity analysis. The low standard deviation (0.64 pp across 3 seeds) effectively addresses my concern regarding the robustness of the LLM-generated counterfactuals and confirms that the detected biases are not merely artifacts of the generation process.
> >
> > Additionally, the authors' justification for focusing on unverbalized biases, specifically its application for systems relying on Chain-of-Thought (CoT) monitoring for safety oversight, and the pipeline's ability to discover non-predefined biases, is well-reasoned. I also appreciate the inclusion of the discussion distinguishing this work from concurrent methods like BiasScope and clarifying the methodological extensions over Atanasova et al. (2023).
> >
> > However, while my technical and motivational concerns have been largely resolved, I still retain reservations regarding the core methodological novelty of the work. As the authors acknowledge, the framework is heavily modular and relies on a combination of existing paradigms: LLM-based concept brainstorming, LLM-based counterfactual generation, and standard statistical significance testing. While filtering specifically for CoT omission is a highly practical and useful application for auditing, the underlying algorithmic innovation of the pipeline itself remains incremental.
> >
> > Because the contribution relies more on the synthesis and practical application of existing techniques rather than introducing a fundamentally novel methodology, I will keep my score at a 4 (Weak Accept). I still view this as a technically solid, well-executed, and practically useful paper that brings value to the AI safety and fairness community.

---

### Decision · Program_Chairs · 2026-04-30

**Decision:**

Accept (regular)

**Comment:**

Overall reviewers agree that the strengths of the paper are its careful methodology development and sound results. Some of this emerged after the discussion period as authors provided more evidence in support of their claims in response to reviewer request.

Overall the main hesitation of reviewers, and the reason why the recommendation is a weak accept, is that the core conceptual underpinning (of discovery of concepts automatically to test for CoT unverbalized biases) is a little weak and can't address the many "human" concepts that might lead to unverbalized biases. The authors acknowledge this as a limitation. I might add that the formal definition of an unverbalized bias (there's a bias in outcome and the CoT reasoning trace doesn't refer to the concept) is one way of approaching this, but there might be authors, and one reviewer explores this indirectly by asking where the biases uncovered might be coming from and why.

This isn't a dealbreaker for the paper.